# Hydroxide promotes carbon dioxide electroreduction to ethanol on copper via tuning of adsorbed hydrogen

Mingchuan Luo[1,3], Ziyun Wang [1,3], Yuguang C. Li [1,3], Jun Li [1,2], Fengwang Li [1], Yanwei Lum [1], Dae-Hyun Nam [1], Bin Chen[1], Joshua Wicks [1], Aoni Xu[1], Taotao Zhuang[1], Wan Ru Leow[1], Xue Wang [1], Cao-Thang Dinh [1], Ying Wang [1], Yuhang Wang [1], David Sinton [2] & Edward H. Sargent [1*]

Producing liquid fuels such as ethanol from $CO_2$, $H_2O$, and renewable electricity offers a route to store sustainable energy. The search for efficient electrocatalysts for the $CO_2$ reduction reaction relies on tuning the adsorption strength of carbonaceous intermediates. Here, we report a complementary approach in which we utilize hydroxide and oxide doping of a catalyst surface to tune the adsorbed hydrogen on Cu. Density functional theory studies indicate that this doping accelerates water dissociation and changes the hydrogen adsorption energy on Cu. We synthesize and investigate a suite of metal-hydroxide-interface-doped-Cu catalysts, and find that the most efficient, $Ce(OH)_x$-doped-Cu, exhibits an ethanol Faradaic efficiency of 43% and a partial current density of 128 mA cm$^{-2}$. Mechanistic studies, wherein we combine investigation of hydrogen evolution performance with the results of operando Raman spectroscopy, show that adsorbed hydrogen hydrogenates surface *HCCOH, a key intermediate whose fate determines branching to ethanol versus ethylene.

---

[1] Department of Electrical and Computer Engineering, University of Toronto, Toronto, Ontario M5S 1A4, Canada. [2] Department of Mechanical and Industrial Engineering, University of Toronto, Toronto, Ontario M5S 3G8, Canada. [3]These authors contributed equally: Mingchuan Luo, Ziyun Wang, Yuguang C. Li. *email: ted.sargent@utoronto.ca

The electrochemical generation of $C_{2+}$ fuels and chemicals from $CO_2$ and $H_2O$ enables the storage of intermittent renewable energy[1–6]. Substantial progress has been made in producing gaseous ethylene from the $CO_2$ reduction reaction ($CO_2$RR), and the Faradaic efficiency (FE) now exceeds 70% at an overpotential of 0.55 V (ref. [7]).

By contrast, the electrochemical conversion of $CO_2$ to liquid ethanol — a promising renewable fuel with high energy density and compatibility with existing storage and transportation infrastructure — has seen more limited progress thus far. Indeed, today's best $CO_2$ electrocatalysts fail to provide majority ethanol production, instead preferring ethylene[8–11].

The optimization of intermediate binding energetics provides a framework in which to evaluate and design for desired electrocatalytic performance[12,13]. For $CO_2$RR specifically, the binding energy of CO ($\Delta E_{CO}$) is an important descriptor that has enabled the prediction of a number of promising $CO_2$RR candidates[14]. Experimental studies have similarly followed the correlation between adsorbed CO ($CO_{ad}$) as a function of alloying/doping with elements, including Zn (ref. [15]), Ag (refs [16–18]), Au (ref. [19]), S (ref. [20]), B (ref. [21]), and N (ref. [22]), as well as with engineering of facets[23] and morphology[24–26].

In spite of these impressive efforts, the FE of $CO_2$-to-ethanol remains below 25% if one focuses on studies that achieve commercially-relevant current densities (>100 mA cm$^{-2}$). Even if one includes results down to 6 mA cm$^{-2}$, it has reached only 29% FE.

The scaling relationships among the carbonaceous intermediates[27] along the multi-step reduction pathway to ethanol mean that it is difficult — if only a single site, and thus one degree of freedom, is relied upon to engineer catalyst adsorption energies— to optimize $\Delta E_{CO}$ simultaneously with the initial $CO_2$ adsorption; as well as to optimize the site for ensuing carbon–carbon coupling; and to optimize also the subsequent hydrogenation step. The formation of ethanol with high FE will rely on accessing experimental degrees of freedom that engineer these steps.

We reasoned that, since hydrogen ($H_{ad}$) co-exists with carbon-based intermediates during $CO_2$RR, controlling its presence could potentially offer a new handle to help break the scaling relations. There exist hints at this possibility in the prior literature, such as in the proposed ethylene pathway in which hot water hydrogenates adsorbed *HCCOH, the penultimate reaction intermediate for both ethanol and ethylene[28]. Experimentally, the direct involvement of water in producing ethanol from $CO_2$RR was clarified in a recent isotopic study[29].

We, therefore, pursued means to activate near-surface water molecules with the goal of boosting the production of ethanol.

Our thinking was that cleaving the Cu–C bond of adsorbed *HCCOH could thereby be promoted, favoring thereby the electroproduction of ethanol.

We begin with an investigation of how surface $H_{ad}$ affects the selectivity of $CO_2$RR on Cu. To this end, we designed a catalytic system that allows us to construct both hydroxide- and oxide-doped Cu having tunable surface $H_{ad}$ coverage. DFT studies reveal that this new catalyst facilitates water dissociation and favors $H_{ad}$ formation. We then synthesize Pourbaix-stable hydroxide-doped and oxide-doped Cu catalysts and investigate them both ex situ and in situ. In all cases, we achieve a notable increase in the ratio of ethanol to ethylene production, documenting fully a doubling on the doped-Cu catalysts compared to Cu.

The best of these, Ce(OH)$_x$-doped-Cu, reaches a FE of 43% for ethanol at an operating current density of 300 mA cm$^{-2}$. Mechanistic studies indicate that surface $H_{ad}$ favors the ethanol pathway over ethylene.

## Results

**DFT investigations of the effect of $H_{ad}$.** Since OH$^-$ plays a beneficial role in promoting carbon–carbon coupling[7,30], $CO_2$RR electrocatalysis is carried out today in neutral or alkaline aqueous environment. In this environment, $H_2O$ molecules serve as the proton source for $CO_2$RR.

Cleavage of the H–OH bond is needed to form $H_{ad}$ on catalytic surface — the Volmer step in the hydrogen evolution reaction, HER. This accounts for the slower HER rate in alkaline and neutral media relative to that in acidic media[31]. Previous studies have demonstrated that the introduction of hydroxides or oxides increases $H_{ad}$ coverage by accelerating the water dissociation step[32,33]. The approach tunes $H_{ad}$ without the need to modify the bulk pH.

We thus reasoned that doping Cu with a stable hydroxide or oxide could enhance the surface $H_{ad}$; yet allow us to maintain the alkaline environment that favors carbon–carbon coupling. We carried out DFT calculations on Ce oxide- and Mn oxide-doped-Cu(111) (see the Methods section for details). Ce and Mn oxides were chosen due to their Pourbaix-stability under the reducing potentials used in $CO_2$RR[34]. We examined the water dissociation energy and the H adsorption energy ($E_H$) on both bare and doped-Cu(111). Figure 1 and Supplementary Fig. 1 show that water dissociation is more favorable on the oxide/hydroxide-doped-Cu surface (by 0.48 eV and 0.39 eV for Ce and Mn oxide, respectively) in comparison with that on the pure Cu. The adsorption of hydrogen is also stabilized on the oxide-doped-Cu surface. $E_H$ is more favorable on Mn oxide/Cu than that of Ce oxide/Cu, which suggests that doping metal oxides on Cu

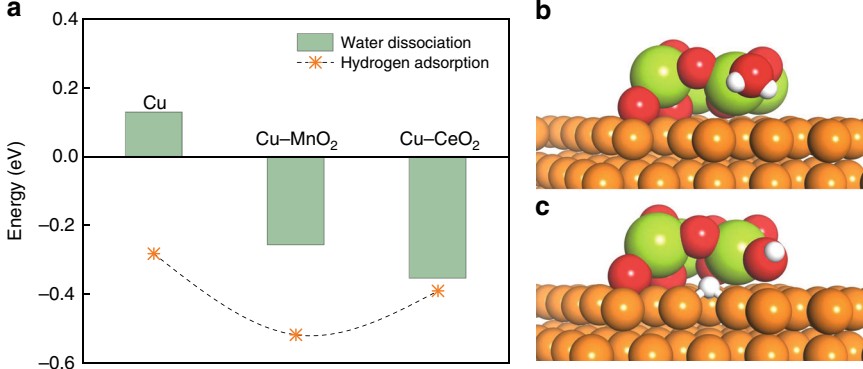

**Fig. 1 Water activation on oxide-modified Cu surfaces. a** Calculated water dissociation reaction energies and hydrogen adsorption energies on various surfaces. **b** Surface configurations of CeO$_2$/Cu with and **c** without adsorbed hydrogen.

provides the mean to control the extent of local $H_{ad}$. These findings suggested that building such hybrid catalysts could enable us to investigate and exploit $H_{ad}$ in $CO_2RR$.

**Catalyst synthesis and characterization.** To synthesize the hybrid catalysts, we began with Cu-sputtered-polytetrafluoroethylene (Cu/PTFE) as the substrate[7], and we deposited either hydroxides or oxides via electrochemical or sputtering methods (Methods). The electrochemical deposition of metallic hydroxide was carried out in a neutral electrolyte containing the corresponding metallic nitrate as the precursor[35] (Supplementary Fig. 2). A cathodic current was first applied to the Cu/PTFE electrode to generate $OH^-$ from nitrate reduction. The metallic hydroxide was then deposited onto the Cu surface via chemical reaction between the metallic cation and locally-generated $OH^-$. During the electrochemical deposition, the color of the Cu surface turned to brown.

To understand the nature of the doped hydroxides, we carried out characterization of the cerium hydroxide-doped-Cu/PTFE sample (denoted $Ce(OH)_x$/Cu/PTFE) using scanning electron microscopy (SEM), transmission electron microscopy (TEM) and scanning transmission electron microscopy (STEM), as well as X-ray diffraction (XRD) and X-ray photoelectron spectroscopy (XPS). Figure 2a shows the typical 3-dimensional networked structure after $Ce(OH)_x$ (Fig. 2a) electrochemically deposited into Cu/PTFE fibers. We propose that this structure facilitates $CO_2$ gas penetration to the triple-phase reaction region. STEM elemental mapping shows a homogeneous distribution of Cu and Ce throughout a single fiber (Fig. 2b). High-magnification SEM images of $Ce(OH)_x$/Cu/PTFE further reveal substantially uniformly-decorated nano-islands on the surface (Fig. 2c) with an average size of 18 nm and a typical range of 6–30 nm (Supplementary Fig. 3). High-resolution TEM reveals the interface between $Ce(OH)_x$ and Cu/PTFE (Fig. 2d). The

corresponding Fast Fourier Transform (FFT, the inset of Fig. 2d) pattern matches that of $Cu_2O$ (111), indicating that the Cu/PTFE-based sample was partially oxidized. The oxidation of Cu is also seen in the XRD patterns of Cu/PTFE (Supplementary Fig. 4a), in which the diffraction peaks corresponding to both Cu and $Cu_2O$ are observed.

The absence of observable lattice spacings in HRTEM, and a corresponding lack of crystalline peaks in XRD, suggest that the electrochemically-deposited $Ce(OH)_x$ exists in an amorphous structure, in agreement with the previous reports[35]. High-resolution XPS spectra for the Cu *2p* region further show the co-existence of both metallic and oxidized states (Supplementary Fig. 4b). The Ce *3d* spectra shows the co-existence of $Ce^{4+}$ and $Ce^{3+}$, indicating that the deposition of cerium species (Supplementary Fig. 4c, d) was indeed achieved. The O *1s* spectra confirm that the cerium species exist as hydroxide (Supplementary Fig. 4e).

Since the chemical states of metals are dependent on the applied potential[36], we carried out operando X-ray adsorption spectroscopy (XAS) to monitor the oxidation states of Cu and Ce during $CO_2RR$ electrocatalysis by looking into the Cu K-edge and Ce $L_3$-edge, respectively. We found that — in agreement with the XPS results — Cu species were slightly oxidized before the reaction (Fig. 2e, Supplementary Fig. 5). However, once a negative potential had been applied during $CO_2RR$, only peaks corresponding to metallic Cu were observed (Supplementary Fig. 5). No change of Cu local structure (i.e., oxidation state, coordination number, and bond distance) was observed throughout $CO_2RR$ process[37] (Supplementary Fig. 6 and Supplementary Table 1). Fig. 2f shows that, once a potential of −0.57 V vs. RHE was applied, the chemical state of Ce underwent an initial reduction. The ratio of $Ce^{3+}$/$Ce^{4+}$ slightly increased from −0.57 V to −0.64 V vs. RHE, after which it remained unchanged upon further-increased reducing potentials. We conclude that an interface is

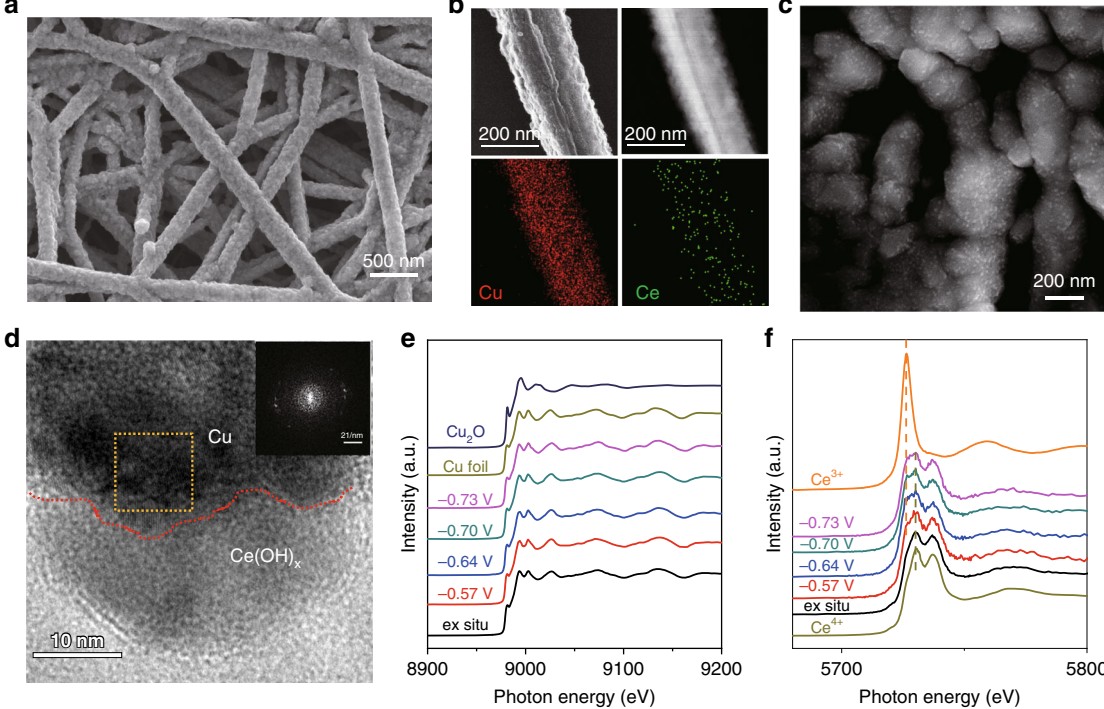

**Fig. 2 Structural characterization of $Ce(OH)_x$ modified Cu catalysts. a** Scanning electron microscope image, **b** STEM image and corresponding EDX mapping for Cu and Ce, **c** High-magnitude SEM image, **d** High-resolution transmission electron microscopy image of $Ce(OH)_x$/Cu/PTFE. The red dashed line draws attention to the interface, and the inset shows the FFT pattern corresponding to the yellow square. **e** Operando Cu K-edge and **f** operando Ce $L_3$-edge XAS of $Ce(OH)_x$/Cu/PTFE catalyst under a number of operating potentials in a flow cell.

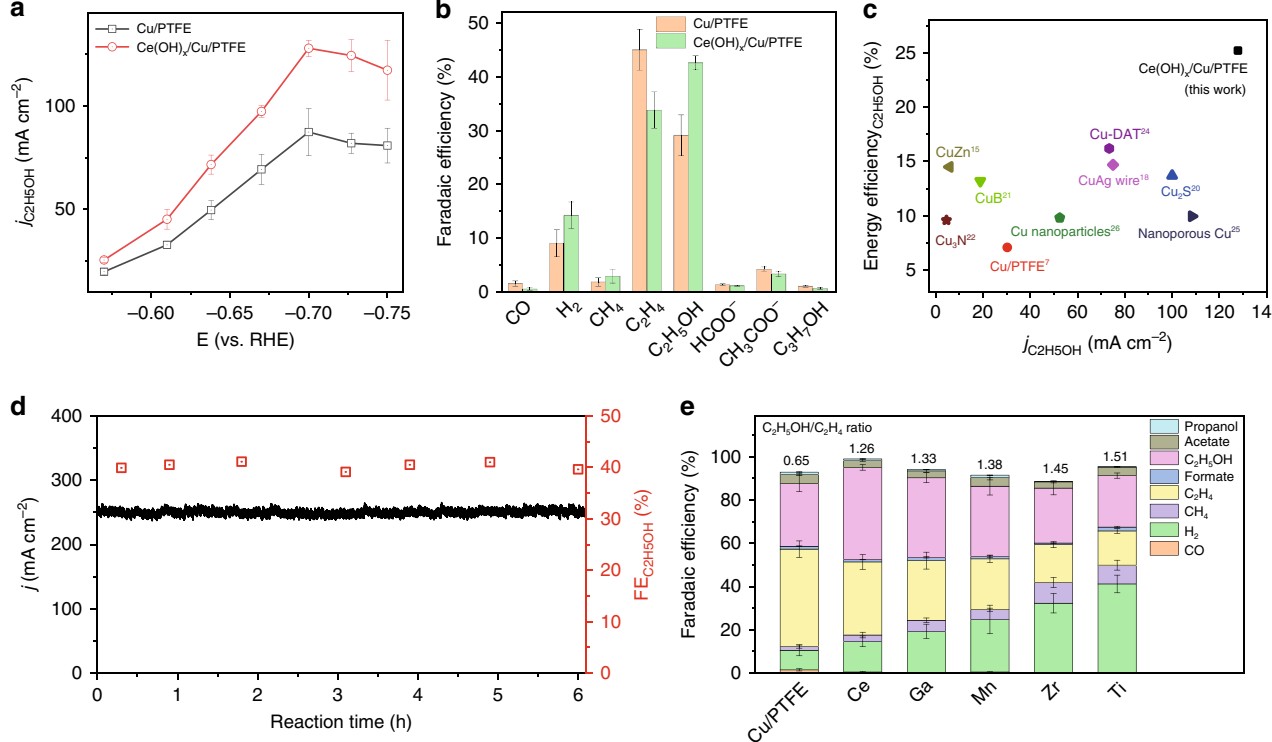

**Fig. 3 Carbon dioxide electroreduction performance. a** Partial ethanol current density of Ce(OH)$_x$/Cu/PTFE and bare Cu/PTFE under various potentials. **b** Product distribution of Ce(OH)$_x$/Cu/PTFE and bare Cu/PTFE at the −0.7 V versus RHE. **c** Energy efficiency as a function of partial current density on Ce(OH)$_x$/Cu/PTFE, in comparison with other reports with operational current density higher than 10 mA cm$^{-2}$. **d** i–t curve (left axis) of Ce(OH)$_x$/Cu/PTFE catalyst along with corresponding Faradaic efficiency of ethanol (right axis). **e** Product distribution of various hydroxides/oxides modified Cu/PTFE electrode, along with corresponding C$_2$H$_5$OH/C$_2$H$_4$ ratio. The error bars represent the standard deviation from at least three independent tests.

provided between metallic Cu and oxidized Ce under the reducing conditions applied during CO$_2$RR electrocatalysis.

**CO$_2$RR performance**. Having established the structural properties of the Ce(OH)$_x$/Cu/PTFE catalyst, we then assessed its CO$_2$RR performance. We used a flow cell set-up with 1 M KOH solution as the electrolyte and throughout included controls involving bare Cu/PTFE (Supplementary Fig. 7 and Supplementary Tables 2–5).

The FE for ethanol reached 43% when the Ce(OH)$_x$/Cu/PTFE catalyst was employed — well above the value of 29% for the Cu control. The Ce(OH)$_x$/Cu/PTFE catalyst also achieved an impressive partial current density, 128 mA cm$^{-2}$ (Fig. 3a), for ethanol, compared to 87 mA cm$^{-2}$ for the copper control. By comparing the product distributions at the optimal potentials, we found that Ce(OH)$_x$ doping had increased the FE toward H$_2$ by 5% compared to the Cu/PTFE baseline (Fig. 3b). This is consistent with DFT results that indicate that surface H$_{ad}$ is enhanced via accelerated water dissociation and optimized hydrogen adsorption. The ethanol:ethylene ratio increased from 0.65 (Cu/PTFE) to 1.26 (Ce(OH)$_x$/Cu/PTFE), which is the highest among electrocatalysts that achieve a current density of >6 mA cm$^{-2}$ (Supplementary Table 2). The energy efficiency as a function of partial current density for ethanol (Fig. 3c) reveals that the Ce(OH)$_x$/Cu/PTFE achieved an energy efficiency of 25%. This is, by a factor of 1.6, the highest reported for systems operating above 10 mA cm$^{-2}$ (Supplementary Table 6).

We also evaluated operating stability of the Ce(OH)$_x$/Cu/PTFE catalyst. It provided stable operation over an initial 6 h at current density 250 mA cm$^{-2}$ (Fig. 3d). TEM and SEM images of Ce(OH)$_x$/Cu/PTFE electrode after reaction showed the preservation of the hydroxide/Cu interface, as well as of the well-dispersed Ce

(OH)$_x$ nano-islands on the sputtered Cu surface (Supplementary Fig. 8).

We used labeled $^{13}$CO$_2$ and confirmed that the ethanol was produced from CO$_2$. This test indicates that ethanol contamination is not a source of artefactual ethanol (Supplementary Fig. 9a). The small changes (within 3%) in electrochemically active surface area (ECSA) of the Cu/PTFE before and after the deposition of Ce(OH)$_x$ also excluded the influence of surface area differences on electrocatalytic performance (Supplementary Fig. 9b–d). Due to its electrical insulation, the PTFE substrate is not expected to affect ECSA measurements.

To investigate whether the materials design strategy herein offers a general way to tune CO$_2$RR selectivity, we further tested other stable hydroxide- and oxide-doped Cu catalysts including Ga(OH)$_3$, Mn(OH)$_3$, Zr(OH)$_4$ and TiO$_2$ (ref. [34]). A suite of microscopy and spectroscopy analysis confirmed they are structurally analogous with Ce(OH)$_x$/Cu/PTFE (Supplementary Figs. 10–13). Similar electrocatalytic behaviors were observed in the doped samples: compared to Cu/PTFE controls, H$_2$ and CH$_4$ production increased, and C$_2$H$_4$ decreased (Fig. 3e). A positive correlation was observed between the FE ratio of ethanol/ethylene and the FE of H$_2$ (FE$_{H2}$), with TiO$_2$/Cu/PTFE exhibiting the highest ethanol/ethylene ratio of 1.51. This agrees with the hypothesis that enhanced H$_{ad}$ promotes ethanol over ethylene. The HER activities of the samples evaluated using the same flow cell system in Ar atmosphere showed the same trend as the FE$_{H2}$ during CO$_2$RR; i.e., TiO$_2$/Cu/PTFE > Zr(OH)$_4$/Cu/PTFE > Mn(OH)$_3$/Cu/PTFE > Ga(OH)$_3$/Cu/PTFE > Ce(OH)$_x$/Cu/PTFE, further confirming the enhanced H$_{ad}$ on Cu due to an accelerated Volmer step (Supplementary Fig. 14).

**Mechanistic studies**. We then sought further mechanistic insight into the selectivity of CO$_2$RR. Goddard and co-workers[28,38] have

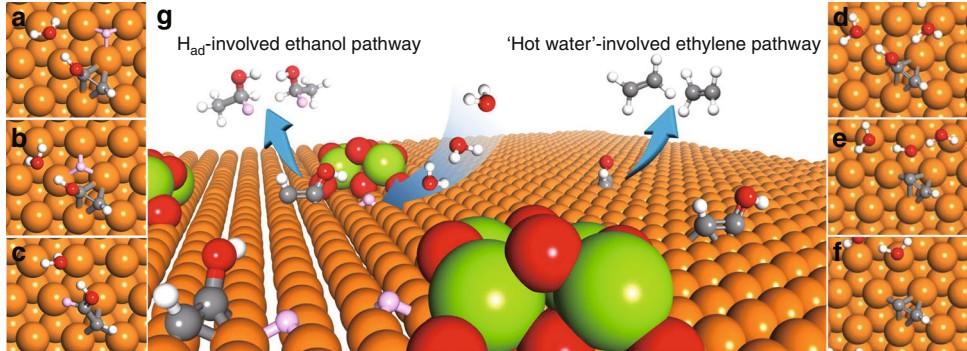

**Fig. 4 Density functional theory calculations on the ethylene and ethanol pathways.** Top views of geometries **a** initial state, **b** transition state, and **c** final state of key reaction towards ethanol, and **d** initial state, **e** transition state, and **f** final state of key reaction towards ethylene. Red, white, gray and orange balls stand for oxygen, hydrogen, carbon, and copper, respectively, while pink balls stand for $H_{ad}$ on Cu.

previously shown that a key intermediate in the branching of ethylene vs. ethanol is *HCCOH (Fig. 4a, d). The ethylene pathway was proposed to be related to the removal of OH in *HCCOH (Fig. 4e) form *CCH (Fig. 4f); from which *CCH is then further hydrogenated, generating ethylene. In contradistinction, *HCCOH is hydrogenated into *HCCHOH (Fig. 4a–c) in the ethanol pathway.

As seen in Fig. 4d–f, surface water molecules are involved in the removal of OH: the hydroxyl group in *HCCOH is surrounded by five other water molecules with hydrogen bonds. In the transition state, the O–C bond between the hydroxyl group and *CCH dissociates with the help of surface water. In the final state, OH is stabilized by water and *CCH is formed. Thus, surface water plays an important role in the ethylene pathway. In the ethanol pathway, the $H_{ad}$ attacks the *HCCOH (Fig. 4b), forming *HCCHOH, the key intermediate towards ethanol. $H_{ad}$ is only involved in the branching reaction towards ethanol. When we enhance $H_{ad}$ coverage, ethanol selectivity is enhanced (Fig. 4g).

To probe experimentally whether hydroxide modification also impacts the adsorption of carbonaceous intermediates on Cu, we carried out in situ Raman measurements and compared bare Cu/PTFE with $Ce(OH)_x$/Cu/PTFE across the potential region −0.24 to −0.73 V under $CO_2RR$ (Supplementary Fig. 15). Due to their short life time, we are unable to provide direct experimental evidence for the *HCCOH intermediates; however, we found negligible influence of $Ce(OH)_x$ on adsorbed CO ($CO_{ad}$) — the Raman shift of frustrated rotation, and stretching, associated with Cu–CO, remained in the same position after $Ce(OH)_x$-modification of the Cu surface. Given the scaling relationship between $CO_{ad}$ and other carbonaceous intermediates[14], we deduced that the electrocatalytic differences between bare Cu/PTFE and $Ce(OH)_x$/Cu/PTFE were unlikely to have originated from changes in the adsorption of carbonaceous species.

It is worth noting that the hydroxide deposition on Cu also promotes the $CH_4$ production from $CO_2RR$. Buonsanti and co-workers[39] recently reported the colloidal synthesis of a class of $Cu/CeO_{2-x}$ heterodimers that showed a $CO_2$-to-$CH_4$ FE of 54% in $KHCO_3$ solution, exceeding the physically-mixed and individual controls. With the aid of DFT studies, they assigned the enhanced $CH_4$ production to the interface comprised of Cu, Ce, and O-vacancy sites that enabled breaking of the $CHO^*/CO^*$ scaling relation. This mechanism investigated herein may contain analogies with how the hydroxide/Cu interface promotes $CH_4$ production through the C1 pathway.

## Discussion

In summary, we reported an approach to higher-efficiency $CO_2$-to-ethanol conversion levering tuning of the adsorption of

hydrogen on Cu. The cerium hydroxide-doped copper catalyst provided a 43% FE at a total current density of 300 mA cm$^{−2}$. Mechanistic studies indicated that $H_{ad}$ on Cu favors the ethanol over the ethylene pathway by attacking the Cu–C bond of the *HCCOH intermediate. The findings suggest further avenues to engineer hybrid catalysts that contribute multiple degrees of freedom to the design of multi-step $CO_2$ reduction reactions.

## Methods

**Electrode preparation.** Cu/PTFE electrodes were prepared by sputtering a Cu layer of 300 nm in thickness onto a PTFE membrane (average pore size of 450 nm) using a Cu target (99.99%) at a rate of 1 Å s$^{−1}$.

Using the Cu/PTFE as the substrate, we electrochemically deposited various hydroxides in a three-electrode electrochemical cell. A potentiostat (Metrohm-Autolab, PGSTAT204) was used for the electrodeposition. The Cu/PTFE, a platinum foil and an Ag/AgCl electrode (saturated with KCl) were used as the working, counter and reference electrodes, respectively. The electrodeposition solution comprised 0.1 M KCl as the supporting electrolyte, and 0.025 M corresponding nitrate salts (cerium nitrate, 99.99%, Sigma-Aldrich; gallium nitrate, 99.9%, Sigma-Aldrich; zirconium oxynitrate, 99.99%, Sigma-Aldrich; manganese nitrate, 99.99%, Sigma-Aldrich) as the precursor. A current density of −0.5 mA cm$^{−2}$ was held for a defined length of time (10, 20, 30, 40, and 50 min) to achieve varied surface coverage of hydroxides. Following the completion of the deposition, the working electrode was rinsed with DI water for at least three times and subsequently dried in $N_2$ atmosphere. Due to the instability of titanium nitrate, we deposited titanium oxides onto Cu/PTFE via a sequential sputtering of Cu followed by a layer of $TiO_2$ (5 nm).

**Materials characterization.** The morphology of the electrodes was characterized using scanning electron microscopy (SEM, Hitachi S-5200) with a 5-kV beam voltage. Transmission electron microscopy (TEM) and elemental mapping images were collected using a Hitachi HF-3300, at an acceleration voltage of 300 kV, equipped with a Bruker energy dispersive X-ray spectroscopy (EDX) detector. The acquisition time in the EDX studies was 3 min. Powder X-ray diffraction (XRD) patterns were recorded using a Bruker D8 using Cu-Kα radiation (λ = 0.15406 nm). X-ray photoelectron spectroscopy (XPS) was conducted on a PHI 5700 ESCA System using Al Kα X-ray radiation (1486.6 eV) for excitation. Operando X-ray absorption spectroscopy (XAS) investigations were carried out at the 9BM beamline of the Advanced Photon Source (APS) located in the Argonne National Laboratory (Lemont, IL). Detailed information regarding *operando* XAS tests in flow cells is available in a previous report[37]. $Cu_2O$, Cu foil, cerium oxide, and cerium oxalate hydrate were used as the reference samples. In situ Raman measurements were performed on a Renishaw inVia Raman Microscope in a modified flow cell and a water immersion objective (×63 ) with a 785 nm laser, using a 5 s integration and averaging 20 scans per region. In the above systems, platinum wire and an Ag/AgCl electrode were used as the counter and reference electrode, respectively.

**Electrochemical measurements.** Electrochemical studies were carried out using an electrochemical flow cell consisting of a gas chamber, a cathodic chamber, and an anodic chamber. The PTFE-based working electrode was fixed between the gas and cathodic chambers, with the catalysts layer side facing the cathodic chamber (geometric active surface area of 1 cm$^2$). An anion exchange membrane (Fumasep FAA-3-PK-130) was used to separate the anodic and cathodic chambers. All electrochemical tests were conducted on an Autolab PGSTAT204, with an Ag/AgCl electrode and Ni foam being the reference and counter electrodes, respectively. Potentials were converted to the reversible hydrogen electrode scale after iR

correction. Electrochemical impedance spectroscopy (EIS) in the frequency range of $10^5$–$10^{-1}$ Hz and an amplitude of 10 mV was used to determine the R value.

For performance studies, 1 M KOH was used as the electrolyte, and it was circulated through the cathodic and anodic chambers using peristaltic pumps at a rate of 10 mL min$^{-1}$. The flow rate of $CO_2$ gas through the gas chamber was controlled to be 50 sccm using a digital gas flow controller. Gas chromatography (PerkinElmer Clarus 600) with a flame ionization detector (FID) and a thermal conductivity detector (TCD) was used to analyze the gas products, collected from the end of the gas chamber. $^1H$ NMR spectroscopy (600 MHz, Agilent DD2 NMR Spectrometer) with water suppression was used to analyze the liquid products, using $D_2O$ and DMSO as the lock solvent and internal reference, respectively. The hydrogen evolution reaction (HER) activities of various electrodes were evaluated in the same flow cell system, with the flow gas changed from $CO_2$ to Ar.

**DFT calculations**. All DFT calculations were carried out using the Vienna ab initio simulation program (VASP; https://vasp.at/)[40–43]. The projected augmented wave approach[44,45] was used to describe the electron-ion interactions with cutoff energy at 450 eV. The generalized gradient approximation with the Perdew, Burke and Ernzerhof exchange correlation functional was used[46]. For all the cerium related calculations, due to the strong correlations of the partially filled Ce 4f states, we employed the Hubbard parameter, U, to illustrate the on-site coulombic interaction[47]. A U–J value of 4.5 eV was chosen for Ce according to a previous study[48]. In order to illustrate the long-range dispersion interactions between the adsorbates and catalysts, we employed the D3 correction method by Grimme et al.[49]. Brillouin zone integration was accomplished using a $3 \times 3 \times 1$ Monkhorst-Pack k-point mesh. Four layers of Cu(111) surface was optimized, with the top 2 layers relaxed and bottom 2 layers fixed. Two molecular units of $CeO_2$ were introduced on to the Cu surface and optimized. To keep the model consistent, we replaced Ce atoms with Mn atoms and re-optimized the structure for the Mn oxide calculations. The water dissociation energy was calculated using $E_{dissociation} = E_{H_2O^*} - E_{H*+OH*}$, and the hydrogen adsorption energy was calculated using $E_{H_{ad}} = E_{H^*} + E_{slab} - 0.5E_{H_2}$, where * designates a surface adsorbed specie.

## Data availability
The data that support the findings of this study are available from the corresponding author on reasonable request.

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

## Acknowledgements

The authors acknowledge funding supporting from Suncor Energy, the Ontario Research Fund and the Natural Sciences and Engineering Research Council (NSERC). All DFT calculations were performed on the IBM BlueGene/Q supercomputer with support from the Southern Ontario Smart Computing Innovation Platform (SOSCIP) and Niagara supercomputer at the SciNet HPC Consortium. SOSCIP is funded by the Federal Economic Development Agency of Southern Ontario, the Province of Ontario, IBM Canada Ltd., Ontario Centres of Excellence, Mitacs, and 15 Ontario academic member institutions. SciNet is funded by the Canada Foundation for Innovation, the Government of Ontario, Ontario Research Fund – Research Excellence, and the University of Toronto. This research used synchrotron resources of the Advanced Photon Source (APS), an Office of Science User Facility operated for the U.S. Department of Energy (DOE) Office of Science by Argonne National Laboratory, and was supported by the U.S. DOE under Contract No. DE-AC02-06CH11357, and the Canadian Light Source and its funding partners. The authors thank T. P. Wu, Y. Z. Finfrock, and L. Ma for technical support at 9BM beamline of APS. J.L. acknowledges the Banting Postdoctoral Fellowships program. D.S. acknowledges the NSERC E.W.R Steacie Memorial Fellowship.

## Author contributions

E.H.S. supervised the project. M.L. designed and carried out the experiments. Y.C.L. and Z.W. designed and carried out the DFT calculations. J.W., A.X., T.Z., and D.H.N. performed the XPS and XRD measurements. J.L., D.H.N., and Y.L. performed and analyzed the in situ XAS measurements. M.L. and F.L. performed and analyzed the Raman measurements. B.C. collected the STEM images and did the EDX mapping. W.L., Y.W., and X.W., collected the SEM images and did the EDX analysis. Y.H.W. prepared sputtered metal electrodes. C.T.D., D.S., and E.H.S. edited the paper. All authors discussed the results. M.L., Y.C.L., and Z.W. wrote the paper.

## Competing interests

The authors declare no competing interests.
