## [Peer Review File · Nature Communications]

Reviewers' comments:

Reviewer #1 (Remarks to the Author):

This manuscript by Luo et al. reports the use of Cu-hydroxide/oxide interfaces for enhanced ethanol versus ethylene production from CO₂ electroreduction (CO₂RR), with an ethanol Faradaic efficiency of 43% and a partial current density of 128 mA/cm² acquired in a flow cell with an alkaline KOH electrolyte. The enhanced adsorbed hydrogen at the interface was reported to be the main contributor for the preferred ethanol pathway. It was postulated that the adsorption of hydrogen can be tuned by employing different Cu-hydroxide/oxide interfaces or by tuning the hydroxide coverage. While these findings are interesting, several important issues need to be clarified before considering this manuscript for publication in any rigorous journal. In particular, insufficient analysis of the data provided has been undertaken which makes the validity of the claims made questionable. Further details are given below.

1. The present assignment of Ce⁴⁺ and Ce³⁺ species is wrong (switched), and some features in the spectra ignored. This is a very serious mistake since the reducibility of the interface might play a key role in the selectivity trend observed. Furthermore, the XPS data shown in Fig. S3 should be properly fitted and the Ce³⁺/Ce⁴⁺ ratio shown in the paper.

2. Despite some claims made in the manuscript, the reported ethanol FE is not too impressive as compared to some of the related literature (e.g., *ChemistrySelect*, 2016, 1, 6055–6061), which showed an ethanol FE of 63% acquired in an H-cell with neutral KHCO₃ as electrolyte.

3. Although the authors showed a positive correlation between the ethanol/ethylene FE ratio and the enhanced adsorbed hydrogen, it is not clear whether other carbon-containing intermediates are also affected by the presence of the Cu-hydroxide interface. Can the authors provide any experimental evidence for the key *HCCOH and *HCCHOH intermediates for the ethanol formation by means of an spectroscopic method such as IR or Raman?

4. A different explanation was reported recently for the activity and selectivity of a similar Cu-CeO_x interface (*ACS Catal.*, 2019, 9, 5035-5046), where O-vacancy sites were key, with intermediates binding to both Cu and Ce atoms, what made possible breaking the CHO*/CO* scaling relation. This might explain why methane production generally increased in Fig.3e. The partially reduced oxides (e.g., CeO_x) might also favor the initial adsorption of CO₂ and other intermediates at the interface. The authors should at least describe the arguments against the mechanism proposed in the former article by Buonsanti group. With the data provided here I am not convinced that the hydroxide modification does not also affect the adsorption of carbonaceous intermediates on the Cu or Cu-hydroxide interface.

5. Are these Cu-hydroxide interface structures stable during CO₂RR? Any agglomeration of the deposited nanoparticles? TEM images after reaction should be provided.

6. The authors conducted double layer capacitance measurements on Cu/PTFE and Ce(OH)₄/Cu/PTFE electrodes. However, it should be considered that this will include contributions from all the materials of the electrodes, mostly from the porous PTFE substrate.

7. It would be very important to establish a correlation between the physical and chemical properties of the metal-hydroxide-doped Cu catalysts. In order to achieve this, a more systematically characterization of the samples presented is needed, in particular, in depth analysis of the XPS and XAFS data. As already indicated in Comment 1, the XPS peak assignment is wrong, and no rigorous analysis of the XAFS data is presented either. According to the authors, hydroxide and oxide doping of a catalyst surface change the Had energy of Cu. Do they observe any changes in either the chemical state of Cu or Ce under reaction conditions? Do the structural properties of Cu change? Even though

such information could be extracted at least from their operando XAFS data (the XPS are ex situ and therefore prone to a possible re-oxidation), the author did not provide any detailed analysis of their data. In order for this work to be further considered in a rigorous scientific journal, the data presented should be properly analyzed. For example, a description of the chemical state of the different samples in their as prepared state (Ce(OH)₄/Cu/PTFE and bare Cu/PTFE) and their possible evolution under reaction conditions should be presented, together with structure parameters obtained from the XAFS data.

8. In Figure 2f, the authors found that the chemical state of Ce gradually changed from Ce⁴⁺ to a mixture of Ce⁴⁺ and Ce³⁺. What is the role of Ce³⁺ during CO₂ reduction? Is there a possibility that such species play a role in enhancing ethanol production?

Reviewer #2 (Remarks to the Author):

Copper:metal hydroxide catalysts promote CO₂ electroreduction to ethanol
2 via tuning of adsorbed hydrogen

This paper takes on the challenging problem of dramatically increasing the FE of ethanol while decreasing that for ethylene. Normally FE for ethylene may be 10 higher than ethanol.

However rather than trying to vary conditions experimentally, they used the best current understanding of the mechanism in which HCCOH* is a common intermediate for forming both ethylene and ethanol, normally more than 10:1 in favor of ethylene and they investigated how to favor the branching to ethanol from this common intermediate. Here they consider how to change the relative energy of H bonding to the surface and the dissociation of water.

They use the Pourbaix diagrams to identify oxides that are likely to be stable under electrocatalysis conditions (MnO_x and CeO_x). They did experiments on just these cases. Indeed they found spectacular improvements for CeO_x, with the ratio of ethanol/ethylene increasing from 0.65 to 1.26, which they report. They then examined other metal-oxides and found that TiO₂ increases the ratio to 1.51.

They applied the full spectrum of experimental tools: scanning (SEM), TEM, STEM, XRD, and XPS to characterize the system.

This is an innovative new strategy for improving electrocatalytic performance using theory and mechanism to select very specific choices in the catalyst modifications. Moreover they show experimentally that this the new strategy works spectacularly. Clearly this paper is highly appropriate for Nature Comm. I strongly recommend accepting as it is.

signed: William A. Goddard III

Reviewer #1

General comment: This manuscript by Luo et al. reports the use of Cu-hydroxide/oxide interfaces for enhanced ethanol versus ethylene production from CO₂ electroreduction (CO₂RR), with an ethanol Faradaic efficiency of 43% and a partial current density of 128 mA/cm² acquired in a flow cell with an alkaline KOH electrolyte. The enhanced adsorbed hydrogen at the interface was reported to be the main contributor for the preferred ethanol pathway. It was postulated that the adsorption of hydrogen can be tuned by employing different Cu-hydroxide/oxide interfaces or by tuning the hydroxide coverage. While these findings are interesting, several important issues need to be clarified before considering this manuscript for publication in any rigorous journal. In particular, insufficient analysis of the data provided has been undertaken which makes the validity of the claims made questionable. Further details are given below.

General reply: We thank the reviewer for constructive feedback. Based on these comments, we further improved the work, especially with regards to structural analysis and mechanistic understanding. This is seen in the revised manuscript (highlighted in yellow) and is detailed below.

Comment 1: The present assignment of Ce⁴⁺ and Ce³⁺ species is wrong (switched), and some features in the spectra ignored. This is a very serious mistake since the reducibility of the interface might play a key role in the selectivity trend observed. Furthermore, the XPS data shown in Fig. S3 should be properly fitted and the Ce³⁺/Ce⁴⁺ ratio shown in the paper.

Reply 1: We have now assigned and properly fitted the XPS peaks in light of the work of Buonsanti et al. (*ACS Catal.*, 2019, 9, 5035.). The corrected assignment of Ce⁴⁺ and Ce³⁺ of the Ce 3d XPS spectrum is now shown in Fig. R1a. Additionally, the XPS spectrum (Ce 3d_{5/2}) was fit and is now shown in Fig. R1b. From the fitting results, we found 57% of Ce⁴⁺ and 43% of Ce³⁺ in the cerium hydroxide-deposited-Cu/PTFE; we thus re-denoted the sample as “Ce(OH)_x/Cu/PTFE” instead of “Ce(OH)₄/Cu/PTFE”. We have updated this in the revised manuscript (page 5) and supporting information (Supplementary Fig. 4).

“The Ce 3d spectra shows the co-existence of Ce⁴⁺ and Ce³⁺, indicating that the deposition of cerium hydroxides (Supplementary Fig. 4c and 4d) was indeed achieved.”

Fig. R1. (a) High-resolution X-ray photoelectron spectroscopy of Ce 3d with peak assignments. (b) The experimental and fitting XPS spectra of the Ce 3d_{5/2} in Ce(OH)_x/Cu/PTFE sample. The ratio of Ce⁴⁺ and Ce³⁺ was determined to be 57%:43%.

Comment 2: Despite some claims made in the manuscript, the reported ethanol FE is not too impressive as compared to some of the related literature (e.g., *ChemistrySelect*, 2016, 1, 6055–6061), which showed an ethanol FE of 63% acquired in an H-cell with neutral KHCO₃ as electrolyte.

Reply 2: The above-mentioned paper achieved an ethanol FE of 63% in a H-cell with a current density and an overpotential of approximately 1.75 mA/cm² and 1.29 V, respectively. We are now explicitly clear, every time we mention where the reported faradaic efficiency falls within the literature, that comparisons are made for currents exceeding 10 mA/cm² (Supplementary Table 2).

Comment 3: Although the authors showed a positive correlation between the ethanol/ethylene FE ratio and the enhanced adsorbed hydrogen, it is not clear whether other carbon-containing intermediates are also affected by the presence of the Cu-hydroxide interface. Can the authors provide any experimental evidence for the key *HCCOH and *HCCHOH intermediates for the ethanol formation by means of a spectroscopic method such as IR or Raman?

Reply 3: We investigated more deeply whether the presence of hydroxide affected the carbon-containing intermediate, *CO (*ACS Catal.*, 2017, 7, 7873.). Using in-situ Raman spectroscopy, we observed Raman signals associated with Cu–CO, suggesting that the Cu surface is covered by *CO. We further found that the presence of the hydroxide does not affect the Raman shift of Cu–CO, nor does it change the onset potential (defined as the most positive applied potential) for the observation of Cu–CO signals

(Supplementary Fig. 15). These results suggest that the hydroxide does not change the adsorption energy of the substrate Cu to CO nor does it enrich local CO concentrations.

We now more clearly point out that tracking C_2 intermediates using operando/in-situ spectroscopic techniques has been an ongoing and challenging topic of research (*Nat. Catal.*, 2018, 1, 922.). In light of the reviewer's suggestion, we firstly used the operando Raman spectroscopy to study the $Ce(OH)_x/Cu/PTFE$ electrode under CO_2RR condition; however, we observed no signals associated with C_2 intermediates (*Angew. Chem. Int. Ed.*, 2017, 56, 3621.) We account for this finding through the very short life time of these C_2 intermediates (*ACS Catal.*, 2017, 7, 7873.) We then carried out Raman spectroscopic studies by directly introducing glyoxal into the electrolyte as a probe molecule, which has been suggested by Koper and co-workers to be a key intermediate along the C_2 pathway (*Chem. Sci.*, 2011, 2, 1902.). However, we did not identify a Raman shift related to either glyoxal or its further reduced species (Fig. R2).

In the revised manuscript, we add the following sentence to acknowledge the difficulties in gaining experimental evidence of C_2 intermediates.

*“Due to their short life time, we are unable to provide direct experimental evidence for the $*HCCOH$ intermediates”.*

Fig. R2. Operando Raman spectra of the $Cu/PTFE$, $Ce(OH)_x/Cu/PTFE$ electrodes in flow cell with 1 M phosphate buffer as the electrolyte at -0.45 V versus RHE, after background subtraction.

Comment 4: A different explanation was reported recently for the activity and selectivity of a similar $Cu-CeO_x$ interface (*ACS Catal.*, 2019, 9, 5035-5046), where O-vacancy sites were key, with intermediates binding to both Cu and Ce atoms, what made possible breaking the CHO^*/CO^*

scaling relation. This might explain why methane production generally increased in Fig.3e. The partially reduced oxides (e.g., CeOx) might also favor the initial adsorption of CO₂ and other intermediates at the interface. The authors should at least describe the arguments against the mechanism proposed in the former article by Buonsanti group. With the data provided here I am not convinced that the hydroxide modification does not also affect the adsorption of carbonaceous intermediates on the Cu or Cu-hydroxide interface.

Reply 4: In the paper by the Buonsanti group, the authors developed a novel colloidal seeded-growth synthesis to construct an advanced class of Cu/CeO_{2-x} heterodimers, on which the interface of Cu/CeO_{2-x} was achieved. When used as a CO₂RR electrocatalyst, the heterodimers realized a methane FE of 54% at -1.2 V versus RHE, exceeding the physically-mixed controls and thus validating the beneficial role of the interface. They attributed the enhanced CO₂-to-CH₄ performance to the unique active motif (a combination of Cu, Ce and O-vacancy sites), which breaks the CHO*/CO* scaling relations, as supported by their DFT study.

We do also note these differences: (1) Materials: colloidal Cu/CeO_{2-x} heterodimers in the Buonsanti study *versus* sputtered Cu/PTFE interfacing with Ce(OH)_x in the present study. (2) Evaluation system: mass transport limited H-cell *versus* gas flow cell. (3) Electrolyte: neutral *versus* alkaline.

In light of the reviewer's comment, we cited and discussed this paper in explaining our observation that the CO₂-to-CH₄ performance was also promoted by cerium hydroxide deposition.

"It is worth noting that the hydroxide deposition on Cu also promotes the CH₄ production from CO₂RR. Buonsanti and co-workers³⁹ recently reported the colloidal synthesis of a class of Cu/CeO_{2-x} heterodimers that showed a CO₂-to-CH₄ FE of 54% in KHCO₃ solution, exceeding the physically-mixed and individual controls. With the aid of DFT studies, they assigned the enhanced CH₄ production to the interface comprised of Cu, Ce and O-vacancy sites that enabled breaking of the CHO/CO* scaling relation. This mechanism investigated herein may contain analogies with how the hydroxide/Cu interface promotes CH₄ production through the CI pathway."*

Comment 5: Are these Cu-hydroxide interface structures stable during CO₂RR? Any agglomeration of the deposited nanoparticles? TEM images after reaction should be provided.

Reply 5: We carried out TEM studies of the Ce(OH)_x/Cu/PTFE samples following a 6-h CO₂RR

electrolysis; we confirm the preservation of the Cu-hydroxide interface (highlighted by dash red line in Fig. R3a). To enable a statistical and representative characterization of the $\text{Ce}(\text{OH})_x$ nano-islands on Cu/PTFE, we also carried out SEM characterization on $\text{Ce}(\text{OH})_x/\text{Cu}/\text{PTFE}$ after reaction (Fig. R3b). The deposited $\text{Ce}(\text{OH})_x$ nano-islands were still well-dispersed on the sputtered Cu/PTFE without noticeable agglomeration. The average size of $\text{Ce}(\text{OH})_x$ nano-islands after reaction was determined to be 20 nm (Fig. R3c), only slightly larger than that of the pristine sample (18 nm). Based on these microscopy results, we conclude that these Cu-hydroxide interface structures are stable during CO_2RR electrocatalysis. We have updated the manuscript and supporting information accordingly.

“TEM and SEM images of $\text{Ce}(\text{OH})_x/\text{Cu}/\text{PTFE}$ electrode after reaction showed the preservation of the hydroxide/Cu interface, as well as of the well-dispersed $\text{Ce}(\text{OH})_x$ nano-islands on the sputtered Cu surface (Supplementary Fig. 8)”

Fig. R3. (a) TEM image, (b) SEM image, and (c) corresponding size-distribution histogram of cerium hydroxide nano-islands in the sample after CO_2RR electrolysis.

Comment 6: The authors conducted double layer capacitance measurements on Cu/PTFE and $\text{Ce}(\text{OH})_4/\text{Cu}/\text{PTFE}$ electrodes. However, it should be considered that this will include contributions from all the materials of the electrodes, mostly from the porous PTFE substrate.

Reply 6: In our study, the PTFE substrate is not electrically conductive (unlike the typically-used carbon paper substrate), and thus does not contribute to double layer capacitance. We have clarified this in the revised manuscript on page 6.

Comment 7: It would be very important to establish a correlation between the physical and

chemical properties of the metal-hydroxide-doped Cu catalysts. In order to achieve this, a more systematically characterization of the samples presented is needed, in particular, in depth analysis of the XPS and XAFS data. As already indicated in Comment 1, the XPS peak assignment is wrong, and no rigorous analysis of the XAFS data is presented either. According to the authors, hydroxide and oxide doping of a catalyst surface change the Had energy of Cu. Do they observe any changes in either the chemical state of Cu or Ce under reaction conditions? Do the structural properties of Cu change? Even though such information could be extracted at least from their operando XAFS data (the XPS are ex situ and therefore prone to a possible re-oxidation), the author did not provide any detailed analysis of their data. In order for this work to be further considered in a rigorous scientific journal, the data presented should be properly analyzed. For example, a description of the chemical state of the different samples in their as prepared state (Ce(OH)₄/Cu/PTFE and bare Cu/PTFE) and their possible evolution under reaction conditions should be presented, together with structure parameters obtained from the XAFS data.

Reply 7: We have now provided a more in-depth analysis of the XPS and XAFS data in the revised manuscript. These enable us better to establish better correlation between the physical and chemical properties of the metal-hydroxide-doped Cu catalysts.

To investigate the chemical state of Cu and Ce under reaction conditions, we analyzed the Cu K-edge first derivative spectra and Ce L₃-edge XAS of Ce(OH)_x/Cu/PTFE measured before and during CO₂RR at various potentials (Fig. R4). In agreement with the XPS results (Supplementary Fig. 4b), we found that Cu species were slightly oxidized before reaction. However, once a negative potential had been applied during CO₂RR, only peaks corresponding to those of metallic Cu were observed as indicated in Fig. R4a. The ratio of Ce³⁺/Ce⁴⁺ slightly increased when the potential was changed from -0.57 V to -0.64 V vs. RHE, then it remained unchanged with increased reducing potentials. Therefore, the chemical states of our catalyst would be metallic Cu⁰ interfacing with a mixture of Ce(OH)₃ and Ce(OH)₄.

To monitor the structural evolution of Cu under the reaction conditions, we fit the Cu K-edge EXAFS spectra recorded at R-scale for Cu foil and Ce(OH)_x/Cu/PTFE catalyst before and during CO₂RR at various potentials. As shown in Fig. R5 and Table R1, except for the initial reduction of Cu oxide, no change of Cu local structure (i.e. oxidation state, coordination number and bond distance) was observed throughout the CO₂RR process.

We have added these structural details and rewritten the in-situ XAS part of the revised manuscript.

“We found that – in agreement with the XPS results – Cu species were slightly oxidized before the reaction (Fig. 2e, Supplementary Fig. 5). However, once a negative potential had been applied during CO₂RR, only peaks corresponding to metallic Cu were observed (Supplementary Fig. 5). No change of Cu local structure (i.e. oxidation state, coordination number and bond distance) was observed throughout CO₂RR process (Supplementary Fig. 6 and Table 1).³⁷ Fig. 2f shows that, once a potential of -0.57 V vs. RHE was applied, the chemical state of Ce underwent an initial reduction. The ratio of Ce³⁺/Ce⁴⁺ slightly increased from -0.57 V to -0.64 V vs. RHE, after which it remained unchanged upon further-increased reducing potentials.”

Fig. R4. (a) Cu K-edge First derivative spectra measured before and during CO₂RR at various potentials.

Fig. R5. Fitting of the Cu K-edge EXAFS spectra recorded at R-space for Cu foil (a) and Ce(OH)_x/Cu/PTFE catalyst before (b) and during (c-f) CO₂RR at various potentials.

	Scatter	CN	R(Å)	ΔE (eV)	σ ² (x 10 ⁻³ Å ²)
Cu foil	Cu-Cu	12	2.54(3)	4.5(5)	8.62

Ex-situ	Cu-O	0.2(4)	1.83(6)	8.1(9)	8.47
	Cu-Cu	11.2(9)	2.54(0)		
-0.57 V	Cu-Cu	11.4(5)	2.53(4)	5.1(0)	8.06
-0.64 V	Cu-Cu	11.3(9)	2.53(4)	5.0(3)	8.03
-0.70 V	Cu-Cu	11.5(7)	2.53(5)	5.2(8)	8.10
-0.73 V	Cu-Cu	11.2(5)	2.53(4)	5.5(6)	7.92

Tab. R1. Tabulated fitting results of the Cu K-edge EXAFS spectra for Cu foil and Ce(OH)_x/Cu/PTFE catalyst before and during CO₂RR at various potentials.

Comment 8: In Figure 2f, the authors found that the chemical state of Ce gradually changed from Ce⁴⁺ to a mixture of Ce⁴⁺ and Ce³⁺. What is the role of Ce³⁺ during CO₂ reduction? Is there a possibility that such species play a role in enhancing ethanol production?

Reply 8: In the proposed mechanism, both Ce(IV) and Ce(III) in hydroxides enhance water dissociation and increase the hydrogen adsorption energy (E_H) on Cu, thus generating high surface coverage of H_{ad} and promoting the ethanol pathway over ethylene.

To verify this, we further carried out DFT calculations to study the role of Ce(III) in the water dissociation and the hydrogen adsorption on Cu, similar to what we did for Ce(IV). As can be seen from Fig. R6, the water dissociation energy and hydrogen adsorption energy of Ce(III)-doped Cu are -0.42 and -0.45 eV, respectively, which are close to those of Ce(IV)-doped Cu while stronger than that of bare Cu. We conclude therefore that Ce(IV) and Ce(III) hydroxides provide a similar enhancement of ethanol production in CO₂RR.

Fig. R6. Calculated water dissociation reaction energies and hydrogen adsorption energies on bare Cu, Ce(III)-doped and Ce(IV)-doped Cu surfaces.

Reviewer #2

General comment: Rviewer#2 raised no comment, and suggested the acceptance of this work.

General reply: We thank the reviewer for the recommendation of our work.

Reviewers' comments:

Reviewer #1 (Remarks to the Author):

I appreciate the authors' efforts to revise the manuscript based on the reviewer comments. From the revised manuscript and the rebuttal letter it appears that the authors were able to address some of the original concerns. The more in depth analysis of the XPS and EXAFS data presented now is appropriate.

Nonetheless, the results presented are still confusing and contradictions can still be found. Therefore, I still cannot recommend the revised manuscript for acceptance in a high impact/high reputation journal such as Nature Communications. Below are more detailed comments on some of the remaining issues:

1. The authors used Ce oxide species (CeO_2) in the model of their DFT calculations. However, it is claimed in the experimental text that Ce hydroxide ($\text{Ce}(\text{OH})_4$) species are likely present in their samples. Since it was discussed that depending on the content of O and OH groups on the edge sites of their samples water dissociation and hydrogen adsorption energies could be varied, one would expect that the theoretical model would be a better representation of the "best knowledge" composition/oxidation state of the experimental samples.

2. The authors still need to provide additional characterization of their samples regarding the Ce species. Even though there is no direct evidence that the Ce species are present as hydroxide, the authors have concluded it and used it in their arguments.

3. In Figure 2f, reference spectra for the Ce species are missing (unlike the data included for the Cu K-edge analysis). Such additional information is needed for a proper structural and chemical state analysis of their samples.

4. The authors provide only limited electrochemical data. Additional information should be added before considering publication in any rigorous journal, in particular: (a) LSV data: the authors only include Ar-saturated HER data in suppl. Figure 14. There are no CO_2RR data. (b) F. E. and partial current density of all products. (c) CV data extracted from double-layer capacitance measurements.

5. They authors have performed in situ Raman, but the data shown do not appear to support their claims. Adsorbed CO on Cu was identified, but no clear difference between the Cu and $\text{Cu}/\text{Ce}(\text{OH})_x$ materials as would have been expected. This should be clarified.

6. The authors presented operando EXAFS and Raman data in a flow cell configuration. Such set-up is not standard and it would be useful for the readers if the authors were to include the details concerning their cell design and implementation in the supplementary documents.

7. How large was the Ce loading on these materials? For how long was the EDX shown in figure 2b acquired? The Ce signals are very weak and could be confused with background points.

8. In Supplementary figure 8 the authors should include EDX and Cu/Ce ratio. From the TEM and SEM images one cannot really judge if there is Ce after reaction. Do they have any other complementary technique to quantify the Ce content before and after the reaction?

9. In Supplementary figure 15 it should be indicated at which potential was each spectrum measured.

Reviewer #1

General comment: I appreciate the authors' efforts to revise the manuscript based on the reviewer comments. From the revised manuscript and the rebuttal letter it appears that the authors were able to address some of the original concerns. The more in depth analysis of the XPS and EXAFS data presented now is appropriate. Nonetheless, the results presented are still confusing and contradictions can still be found. Therefore, I still cannot recommend the revised manuscript for acceptance in a high impact/high reputation journal such as Nature Communications. Below are more detailed comments on some of the remaining issues:

General reply: We appreciate the reviewer's comments. We have revised the manuscript accordingly (highlighted by yellow background) and provide a detailed response below.

Comment 1: The authors used Ce oxide species (CeO_2) in the model of their DFT calculations. However, it is claimed in the experimental text that Ce hydroxide ($\text{Ce}(\text{OH})_4$) species are likely present in their samples. Since it was discussed that depending on the content of O and OH groups on the edge sites of their samples water dissociation and hydrogen adsorption energies could be varied, one would expect that the theoretical model would be a better representation of the “best knowledge” composition/oxidation state of the experimental samples.

Reply 1: We now construct a model based on $\text{Ce}(\text{OH})_3$, justified on the basis that Ce^{3+} is observed experimentally to be the preponderant chemical state during CO_2RR electrolysis, as seen in our operando XAS investigations. We studied $\text{Ce}(\text{OH})_3$ on Cu and re-calculated the energies of water dissociation (-0.397 eV) and hydrogen adsorption (-0.876 eV). As shown in Figure R1, both Ce oxides and Ce hydroxide enable enhanced water dissociation and hydrogen adsorption on Cu. We have added a summary of these new studies to the revised Supplementary Figure 1.

Figure R1. Calculated water dissociation reaction energies and hydrogen adsorption energies on bare Cu, CeO₂-doped, Ce₂O₃-doped and Ce(OH)₃-doped Cu surfaces.

Comment 2: The authors still need to provide additional characterization of their samples regarding the Ce species. Even though there is no direct evidence that the Ce species are present as hydroxide, the authors have concluded it and used it in their arguments.

Reply 2: To ascertain the nature of Ce species, we measured the XPS of Cu/PTFE samples before and after the electrochemical deposition. From the high resolution XPS spectra in the O 1s region (Figure R2 and revised Supplementary Fig. 4e), the Cu/PTFE shows only an oxide-characteristic peak at 529.5 eV, resulting from Cu oxide that is formed while exposed to ambient air during the transfer process for XPS (*Science*, 2018, 360: 783.). However, after electrochemical deposition, there emerges another peak of oxygen located at around 532 eV, which corresponds to oxygen in the form of hydroxide (*Nat. Energy*, 2016, 1: 16053.). We conclude that Ce species in the catalysts investigated herein exist as hydroxide. We have added these new results to the revised Supplementary Figure 4e.

Figure R2. High resolution XPS spectra of the O 1s region of Cu/PTFE and Ce(OH)_x/Cu/PTFE.

Comment 3: In Figure 2f, reference spectra for the Ce species are missing (unlike the data included for the Cu K-edge analysis). Such additional information is needed for a proper structural and chemical state analysis of their samples.

Reply 3: We have added reference spectra for Ce species, which includes Ce^{4+} from cerium oxide and Ce^{3+} from the cerium oxalate hydrate (Figure R3). We have added these to the revised Figure 2f.

Figure R3. Operando Ce L_3 -edge XAS of $\text{Ce}(\text{OH})_x/\text{Cu}/\text{PTFE}$ catalyst under various operational potentials in a flow cell.

Comment 4: The authors provide only limited electrochemical data. Additional information should be added before considering publication in any rigorous journal, in particular: (a) LSV data: the authors only include Ar-saturated HER data in suppl. Figure 14. There are no CO_2RR data. (b) F. E. and partial current density of all products. (c) CV data extracted from double-layer capacitance measurements.

Reply 4: (a) We now add CO_2RR polarization curves of Cu/PTFE and $\text{Ce}(\text{OH})_x/\text{Cu}/\text{PTFE}$ (Figure R4 and revised Supplementary Fig. 7d). (b) We now tabulate the FE and partial current density of all products as a function of applied potential for both Cu/PTFE and $\text{Ce}(\text{OH})_x/\text{Cu}/\text{PTFE}$ (Table R1-4 and revised Supplementary Tab. 2-5). (c) We now include CV data used for double-layer capacitance measurements (Figure R5 and revised Supplementary Fig. 9c).

Figure R4. CO₂RR polarization curves for Cu/PTFE and Ce(OH)_x/Cu/PTFE, recorded under the same conditions as for CO₂RR performance evaluation.

Table R1. FEs to various CO₂RR products on Cu/PTFE electrodes as a function of working potential.

E (V vs. RHE _{IR})	CO	H ₂	CH ₄	C ₂ H ₄	Formate	EtOH	Acetate	PrOH
-0.57	6.2 ± 2.7	10.5 ± 1.5	0.4 ± 0.05	41 ± 5	3.3 ± 0.5	19.8 ± 1.5	0.6 ± 0.5	3.7 ± 0.3
-0.61	4.5 ± 1.3	8.8 ± 2	0.7 ± 0.1	44.8 ± 3.1	2.5 ± 0.2	21.9 ± 1.2	1.3 ± 0.2	3.6 ± 0.4
-0.64	3 ± 1.5	6 ± 2.2	1 ± 0.4	52.5 ± 4.5	1.9 ± 0.3	24.8 ± 2.3	2.3 ± 0.5	3.3 ± 0.2
-0.67	2.2 ± 0.8	7.5 ± 1.1	1.3 ± 0.5	48.3 ± 5.5	1.8 ± 0.2	27.7 ± 2.9	2.9 ± 0.4	2.6 ± 0.3
-0.7	1.5 ± 0.5	9 ± 2.5	1.8 ± 0.8	45 ± 3.8	1.3 ± 0.2	29.1 ± 3.8	4.2 ± 0.4	1 ± 0.2
-0.73	1 ± 0.2	11 ± 2.4	3.5 ± 0.6	38.8 ± 4.2	0.8 ± 0.1	23.4 ± 1.4	4.5 ± 0.5	1.2 ± 0.3
-0.75	0.6 ± 0.2	13.4 ± 3.3	7.7 ± 2.9	33.2 ± 3.1	0.5 ± 0.1	20.2 ± 2.1	5.2 ± 0.3	1 ± 0.4

Table R2. Current densities of various CO₂RR products on Cu/PTFE electrodes as a function of working potential.

E (V vs. RHE _{IR})	CO	H ₂	CH ₄	C ₂ H ₄	Formate	EtOH	Acetate	PrOH
-0.57	6.2	10.5	0.4	41	3.3	19.8	0.6	3.7
-0.61	6.75	13.2	1.05	66.75	3.75	32.85	1.95	5.4
-0.64	6	12	2	105	3.8	49.6	4.6	6.6
-0.67	5.5	18.75	3.25	120.75	4.5	69.25	7.25	6.5
-0.7	4.5	27	5.6	135	3.9	87.3	12.6	3
-0.73	3.5	38.5	10.5	135.8	2.8	81.9	15.75	4.2
-0.75	2.4	53.6	30.8	132.8	2	80.8	20.8	4

Table R3. FEs to various CO₂RR products on Ce(OH)_x/Cu/PTFE electrodes as a function of working potential.

E (V vs. RHE _{IR})	CO	H ₂	CH ₄	C ₂ H ₄	Formate	EtOH	Acetate	PrOH
-0.57	3.5 ± 1.9	12.4 ± 2.1	1.2 ± 0.2	31.1 ± 2.1	4.3 ± 1.1	25.5 ± 1.3	0.4 ± 0.1	1.6 ± 0.7
-0.61	2.2 ± 1.3	11 ± 1.8	1.5 ± 0.3	34.5 ± 2.6	2.6 ± 0.9	30.1 ± 3.2	0.5 ± 0.1	1.3 ± 0.4
-0.64	1.4 ± 1	10.1 ± 1.1	1.7 ± 0.8	38.7 ± 3.3	1.5 ± 0.3	35.8 ± 2.3	1.1 ± 0.3	0.9 ± 0.2
-0.67	0.8 ± 0.3	12.8 ± 1.7	2.2 ± 1	36.6 ± 3	1.3 ± 0.4	38.9 ± 1.1	1.9 ± 0.4	0.7 ± 0.2
-0.7	0.5 ± 0.25	14.2 ± 2.5	2.9 ± 1.3	33.8 ± 3.4	1.1 ± 0.1	42.6 ± 1.3	3.3 ± 0.5	0.6 ± 0.2
-0.73	0.3 ± 0.1	20.1 ± 3.4	5.5 ± 2.6	28.9 ± 2.2	0.7 ± 0.2	35.5 ± 2.7	5.4 ± 0.8	0.3 ± 0.1
-0.75	0.2 ± 0.1	24.8 ± 2.8	9.8 ± 3.2	21.8 ± 1.2	0.5 ± 0.1	29.3 ± 3.6	7.9 ± 1.2	0.2 ± 0.05

Table R4. Current densities of various CO₂RR products on Ce(OH)_x/Cu/PTFE electrode as a function of working potential.

E (V vs. RHE _{IR})	CO	H ₂	CH ₄	C ₂ H ₄	Formate	EtOH	Acetate	PrOH
-0.57	3.5	12.4	1.2	31.1	4.3	25.5	0.4	1.6

-0.61	3.3	16.5	2.25	51.75	3.9	45.15	0.75	1.95
-0.64	2.8	20.2	3.4	77.4	3	71.6	2.2	1.8
-0.67	2	32	5.5	91.5	3.25	97.25	4.75	1.75
-0.7	1.5	42.6	8.7	101.4	3.3	127.8	9.9	1.8
-0.73	1.05	70.35	19.25	101.15	2.45	124.25	18.9	1.05
-0.75	0.8	99.2	39.2	87.2	2	117.2	31.6	0.8

Figure R5. (a) CVs of Cu/PTFE and (b) Ce(OH)_x/Cu/PTFE electrodes for double-layer capacitance measurements, recorded in the non-Faradaic potential region at scan rates of 10, 20, 40, 60, 80, 100 mV s⁻¹ using flow cell in 1 M KOH in N₂ atmosphere.

Comment 5: They authors have performed *in situ* Raman, but the data shown do not appear to support their claims. Adsorbed CO on Cu was identified, but no clear difference between the Cu and Cu/Ce(OH)_x materials as would have been expected. This should be clarified.

Reply 5: We now better explain that *operando* Raman results show no clear difference between Cu/PTFE and Ce(OH)_x/Cu/PTFE. Given the scaling relationship between adsorbed CO and other C-containing reaction intermediates, we conclude that the introduction of Ce(OH)_x in the present study has little influence on the adsorption of reaction intermediates. Following the reviewer’s suggestion, we further clarify this argument in the revised manuscript, writing now on p. 8:

“however, we found negligible influence of Ce(OH)_x on adsorbed CO (CO_{ad}) – the Raman shift of both frustrated rotation and stretching associated with Cu–CO remained at the same position after the modification of Ce(OH)_x to Cu surface.”

Comment 6: The authors presented *operando* EXAFS and Raman data in a flow cell configuration. Such set-up is not standard and it would be useful for the readers if the authors were to include the details concerning their cell design and implementation in the supplementary documents.

Reply 6: We now better explain that we performed *operando* EXAFS experiments using a flow cell setup similar to the one used to evaluate CO₂RR performance. Figure R6 shows the photo and scheme of the testing setup. We cite in Methods prior reports that used this setup (*Nat. Commun.*, 2018, 9, 4614).

In situ Raman measurements were carried out using a Renishaw inVia Raman Microscope in a modified flow cell and a water immersion objective (63x) with a 785 nm laser, using a 5 s integration and averaging 5 scans per region (Figure R7 and revised Supplementary Fig. 15c). The spectra were recorded and processed using the Renishaw WiRE (version 4.4) software. A Ag/AgCl electrode was used as the reference electrode and a Pt wire was used as the counter electrode in all measurements.

Figure R6. (a) Photo of operando EXAFS test in custom designed flow cell configuration. (b) Explosive schematic view of flow cell and enlarged view of GDE for operando hXAS measurement.

Figure R7. Schematic illustration of the electrochemical cell for operando Raman measurements. A water immersion objective and a 785 nm laser were used. An Ag/AgCl (3M KCl) electrode and a Pt wire were used as the reference and counter electrodes, respectively.

Comment 7: How large was the Ce loading on these materials? For how long was the EDX shown in figure 2b acquired? The Ce signals are very weak and could be confused with background points.

Reply 7: We determined the Ce loading of Ce(OH)_x/Cu/PTFE with both EDX and XPS, and list the Ce/Cu atomic ratio in Table R5. The higher Ce/Cu ratio determined from XPS than that from EDX indicates that the Ce(OH)_x species are mainly located on the surface.

We now explain that the acquisition time in the EDX studies of figure 2b is 3 min. We have added this information in the revised Methods section and here also provide the corresponding EDX spectra (Figure R8 and revised Supplementary Fig. 3f). These show characteristic Ce signals.

Table R5. The Ce/Cu ratio of Ce(OH)_x/Cu/PTFE electrodes before and after 6-h CO₂RR electrolysis, measured from EDX and XPS, respectively.

Samples	Ce / Cu ratio (EDX)	Ce / Cu ratio (XPS)
Ce(OH) _x /Cu/PTFE (as-synthesized)	5.3 / 94.7	29.2 / 70.8
Ce(OH) _x /Cu/PTFE (after 6-h CO ₂ RR)	5.9 / 94.1	26.5 / 73.5

Figure R8. Corresponding EDX spectra of the elemental mapping shown in Figure 2b.

Comment 8: In Supplementary figure 8 the authors should include EDX and Cu/Ce ratio. From the TEM and SEM images one cannot really judge if there is Ce after reaction. Do they have any other complementary technique to quantify the Ce content before and after the reaction?

Reply 8: The EDX and corresponding Cu/Ce ratio of Ce(OH)_x/Cu/PTFE after reaction are now included (Figure R9 and revised Supplementary Fig. 8d). We also carried out XPS on the Ce(OH)_x/Cu/PTFE sample after reaction, and the results show that the Ce(OH)_x species is mainly located on the surface (Table R5).

Figure R9. The EDX spectra of $\text{Ce(OH)}_x/\text{Cu}/\text{PTFE}$ electrode after 6-h CO_2RR electrolysis, indicating a Ce/Cu atomic ratio of 5.9/94.1.

Comment 9: In Supplementary figure 15 it should be indicated at which potential was each spectrum measured.

Reply 9: The potential for each spectrum of operando Raman is now indicated (Figure R10 and revised Supplementary Fig. 15).

Figure R10. (a) Operando Raman spectra of Cu/PTFE and (b) $\text{Ce(OH)}_x/\text{Cu}/\text{PTFE}$ electrode in flow cell with 1 M KOH as the electrolyte at various electrochemical potentials, after background subtraction.

REVIEWERS' COMMENTS:

Reviewer #1 (Remarks to the Author):

In the second revision of this work, the authors have made substantial changes and modifications in response to my comments that have served to clarify their story and better support their conclusions. In particular, a more clear analysis and description of their operando spectroscopy data is available now and additional details on their electrochemical setups.

Overall, the authors have addressed satisfactorily all critical issues that I have raised in my previous referee reports and therefore, I can now recommend the publication of this work in the present form.

Reviewer #1 (Remarks to the Author):

In the second revision of this work, the authors have made substantial changes and modifications in response to my comments that have served to clarify their story and better support their conclusions. In particular, a more clear analysis and description of their operando spectroscopy data is available now and additional details on their electrochemical setups.

Overall, the authors have addressed satisfactorily all critical issues that I have raised in my previous referee reports and therefore, I can now recommend the publication of this work in the present form.

Reply: We thank the referee for a constructive review process.